# Environmental Urban Plan for Failaka Island, Kuwait: A Study in Urban Geomorphology

**Ahmed Hassan [1],\*** , **Muhammad G Almatar [2]** , **Magdy Torab [3]** and **Casey D Allen [4,5]**

1   Kuwait Ministry of Education, Kuwait 50001, Kuwait
2   Department of Geography, College of Social Sciences, Kuwait University, Kuwait 5969, Kuwait; muhammad.almatar@ku.edu.kw
3   Department of Geography, Damanhur University, Damanhur 22511, Egypt; magdytorab@art.dmu.edu.eg
4   Department of Biological and Chemical Sciences, Faculty of Science and Technology, The University of the West Indies, Bridgetown TA22, Barbados; casey.allen@cavehill.uwi.edu
5   Stone Heritage Research Alliance, LLC., Kaysville, UT 84037, USA
\*   Correspondence: ameh812000@gmail.com

**Abstract:** Failaka Island, located in the far east of Kuwait Bay about 20 km from the State of Kuwait's coast, represents a focal point for regional geography and history, including natural wonders and archaeological sites dating to the Bronze, Iron, Hellenistic, Christian and Islamic periods. According to environmental data and in coordination with local authorities to develop an urban plan, the island is set to become the first tourist destination for the State of Kuwait. To achieve the Vision of Kuwait 2035, one of the planning objectives centers on Urban Planning for the Establishment of Environmental Cities that Achieve (UPEECA) environmental sustainability criteria. The article then, aims to propose the environmental urban plan for Failaka Island. Based around Environmental Analytical Hierarchical Processes (EAHP) and using the Field Calculator and ModelBuilder functions in ArcGIS, this research centers on the feasibility of carrying out an urban plan using suitability modeling that incorporates 4 factors and 13 criteria covering the island's ecological and human composition. This study utilizes both remote sensing (Unmanned aerial vehicles **UAVs** for 3D imaging) and field study (ground truthing) to identify changes in land use and land cover—such as using sample analysis of the historical sites and soils for tracing evidence and creating/updating a soil map—and create the first geographic information systems (GIS) database for the island that can lead capable of generating a suitability model.

**Keywords:** urban geomorphology; sustainable environment; suitability model-GIS; environmental analytical hierarchical process (EAHP); Failaka Island

## 1. Introduction

Global urban population exceeded the rural population for the first time in human history in 2007. Since then, the proportion of people living in urban areas has continued growing and it is expected that by 2050 nearly two-thirds of the global population will be urban [1] Accordingly, researchers in all fields related to sustainable development—including geoheritage and urban geomorphology—should strive for urban planning based on assessment of environmental potentials that include on-site assessment and land-use suitability analyses, with the aim of identifying the most appropriate spatial pattern for the future land use in order to achieve a sense of sustainable development. In recent years, land-use suitability analysis has been applied to the assessment of agricultural land [2], for the study and evaluation of urban services planning [3], to determine a solar farm location in the Legionowo District, Poland [4], landscape evaluation and planning [5], regional planning and environmental

impact assessment [6,7], and help with deciding on optimal solar power plant locations in Kuwait [8]. Additionally, these types of analyses have been used to help determine land habitats for animal and plant species [9], as well as identifying potential future land use conflicts in North Central Florida [10].

While suitability analysis focuses on the process and procedures used to establish the appropriateness of a system according to the needs of a stakeholder [3], in most lesser-developed countries, people often construct residential buildings without considering resources for the new residential areas. Therefore, it becomes the government's problem to provide the required resources for these areas. Occasionally, some countries adopt urban planning strategies that are incompatible with the environment, increasing environmental problems that make life difficult for the population.

Site suitability analysis is the process of determining the fitness of a given tract of land for a defined use [11]. To find a suitable site for construction of an amenity or urban area, however, sophisticated analyses that consider critical components, such as technical, environmental, physical, social, and economic factors, are required. Tools for identification, comparison, and analyzing multi-criterion decisions for urban site development require proper planning and management, and often incorporate vital geospatial tools, such as remote sensing, geographic information systems (GIS), global positioning systems (GPS), and analytical hierarchical process (AHP) modeling [12,13]. Based on these facets, this study suggests building Environmental Analytical Hierarchical process (EAHP), centered around an integrated technique utilizing the AHP and GIS, to support the assessment and the selection of suitable areas for urban development on Failaka Island in the state of Kuwait.

## 2. Site Setting and Goals

The second-largest Kuwaiti island, after Boubyan island, Failaka Island is located 20 km off the Kuwait coast between 29°23′ and 29°28′16″ North and 48°16′ and 48°24′10″ East, with a size approximately 13.8 km long and varying from 1.8 to 6.5 km in width, creating an area of approximately 46.36 square km (Figure 1; [14]). Triangular shaped with its base in the west and its head in the southeast, Failaka Island remains relatively flat apart from a small hill approximately nine meters high on its western side [15,16].

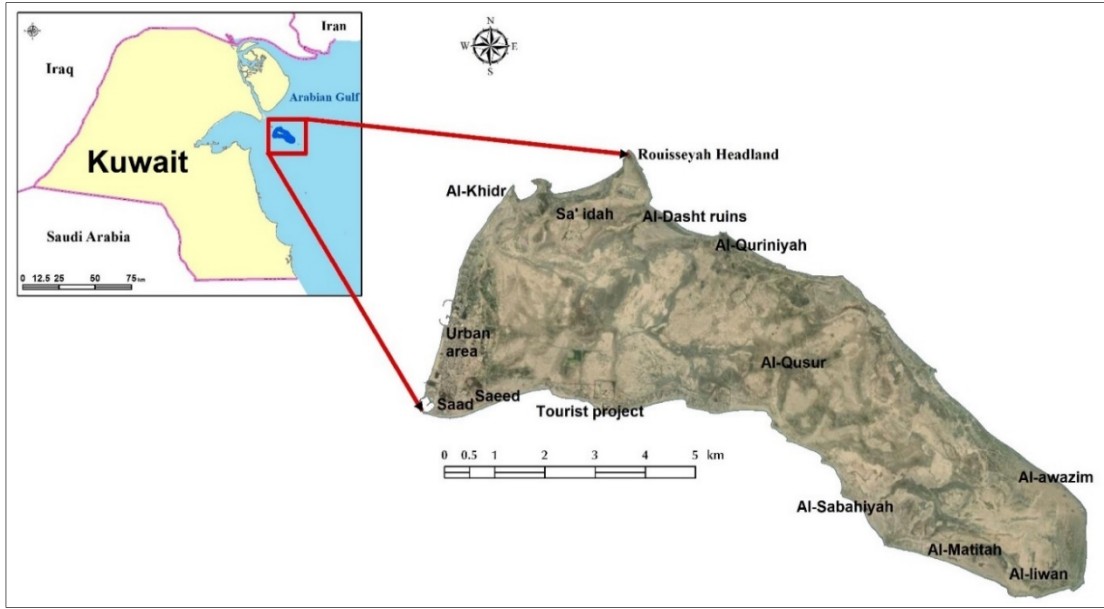

**Figure 1.** Location of the study area, Source: [14].

A focal point of geography and history for Kuwait, Failaka Island offers unspoiled beaches and archaeological sites dating back to the Bronze, Hellenistic, Christian, and Islamic periods. Although the first population figure recorded for Failaka was in 1839 (150 inhabitants), it represents the only

island off Kuwait's coast that has been known to host a population for thousands of years, even increasing during the twentieth century, from 1500 people in 1930 to over 5800 people in 1985 [17]. Failaka Island is considered a unique urban case study as it has had no occupation, and therefore no urbanization, since 1990 when the Iraqi army displaced its residents by forcing them to leave the island. After Kuwait's liberation, the population did not return to the island, and it has been abandoned since that time [17–19]. Given these circumstances, the state and other decision-makers have a unique opportunity to implement potential mitigation and planning strategies for the island—taking into account analyses from specialists in environmental planning, as well as incorporating both human and physical characteristics (i.e., geomorphology, geoarchaeology, soils, coastline types, land use, and geology)—before attempting reconstruction.

To achieve such goals, the government of Kuwait has previously embarked on implementation of various development plans, each of which aims to turn the island into a major tourist destination based on Failaka Island's regional significance. Previous plans concentrated on various development issues, such as a tourist attraction center, residential development projects, and urban infrastructure development. In this context, state plans with differing foci appeared over the past several decades [14,17,19–21], as well as plans conducted by researchers with individual efforts and different scientific backgrounds e.g., [19,22]. While each of these previous studies included specific criteria, none integrate them in a significant way, leading to an incomplete assessment of the island. To remedy these deficiencies, this study proposes a method for environmental planning of Failaka Island by using the Environmental Analytical Hierarchical Process (EAHP) method, incorporated into a geographic information system (GIS), for assessment of multiple factors, including scoring-based scientific literature and local knowledge applied to Multi-Criteria Decision Analysis (MCDA).

## 3. Materials and Methods

Computer modeling has become one of the most important tools in the field of land use and land cover, as GIS and remote sensing are utilized with increasing frequency to investigate landscape changes such as urban expansion [23]. The availability of advanced computer software and tremendous amounts of spatial data encourages GIScientists and remote sensing specialists to implement such technology in modeling land use and land cover classes, and can be a valuable tool for sustainable planning and evaluating social and environmental consequences of landscape fragmentation [24,25]. Given their flexibility, models can also be utilized for studying relationships and interactions amongst system elements and for creating testable hypotheses and predictions regarding patterns and mechanisms [26]. The analyses presented in this article utilize components from various model types and incorporate multiple factors to determine optimal site locations. It bears noting that the plan outlined below takes it cue from often-used criteria-based suitability models.

Cartographic models, for example, are usually analyzed to identify suitable locations based on a set of criteria determined by analysts, such as suitability models—initially specified in qualitative conditions, such as high, medium, and low—but the actual set of criteria is generated according to quantitative measurements required for the ranking process, where the overall suitability ranking process is generated based on the importance of different classes of criteria [27]. For example, Carr and Zwick [10] ranked suitability using values ranging from 1 to 9 for each single suitability layer, where 1 represents the lowest suitability and 9 identifies the highest suitability. These layers were then combined based on land use categories into three main suitability criteria: high, moderate, and low.

Another example of a cartographic model is what Blostad [27] calls weighting among criteria. This process's final step allows for combining all suitability layers into one single layer that determines the best locations. Blostad [27] stated that assigning suitability which depends on non-quantifiable measures is usually difficult, and the process is easier when the importance of the diverse criteria is expressed on a common scale (e.g., 1 to 9 or 1 to 5). Still, there are different kinds of weighting processes that can be used in suitability models to generate the final output. For example, Carr and Zwick [10] used a pairwise comparison methodology called Analytic Hierarchy Process (AHP) to

determine the weight of the layers to produce the final suitability. Similarly, Blostad [27] utilized an importance ranking to assign the weights for slope effects, soil types, and noises cost in his use of a suitability model to determine the best location for home constriction. In the context of Failaka Island, to perform a multi-criteria suitability analysis for the potential environmental urban plan, four main steps must be included: identifying site criteria, data preparation, developing a composite suitability index, and selecting a suitable site (Figure 2).

### 3.1. Identifying Site Criteria

Criteria of land-use suitability for urban development are derived from multi-disciplinary scientific theories related to physical, socio-economical, and ecological attributes [28]. Often, planners and decision makers use only socio-economic parameters in urban planning, and rarely consider environmental aspects. The criteria in Figure 2 can be classified into two main categories: Boolean (exclusionary) and favorable criteria (inclusionary). While Boolean criteria assist in identifying suitability for a particular purpose in a binary manner, favorable criteria reveal this suitability in gradual levels [4,8].

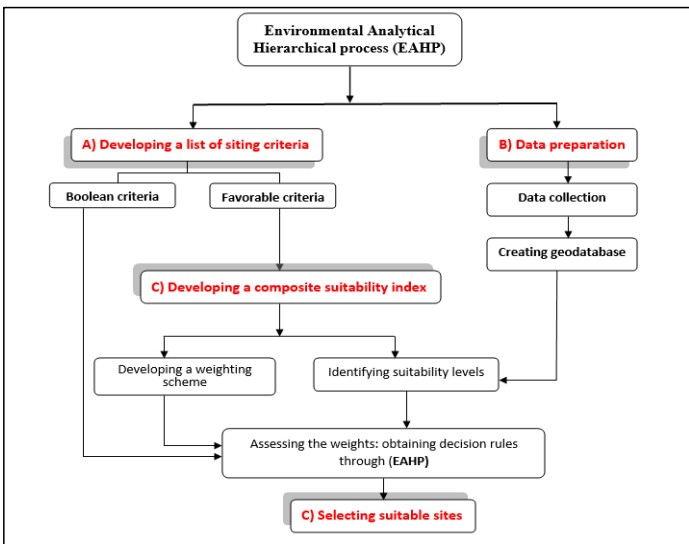

**Figure 2.** Environmental Analytical Hierarchical process (EAHP) Framework. The four steps involved in calculating the (EAHP) are denoted by letters A, B, C, and D.

### 3.2. Data Preparation

Data preparation represents an important step in land suitability analysis [4,6]. Upon identifying analysis criteria, datasets of various indicators were collected. These datasets included:

- A digital elevation model (DEM) from an aerial survey with a spatial resolution of 25 cm, including several topographical analyses such as slope, aspect, and creation of a contour map [14].
- A geomorphological map, derived from various sources (i.e. [14], (ground truthing), and aerial photography by a Mavic2 pro drone UVA.
- Soil analysis—including pH, cations, and anions—for 34 samples covering most parts of the island, leading to an updated soil map for the island, and analyses in the Faculty of Science Laboratory, at Alexandria University (Egypt).
- Coastline types (as a shapefile, from [15].
- Land use/land cover. This represents perhaps the most important dataset for urban and environmental planning, and the most fundamental data of the evaluation model for urban development suitability. It can be derived from multiple sources [14]. To obtain data on the history

of urbanization on the island in the past century, the study relied on several historical maps from Mohamed [16,21].

- A 1:50,000 scale geological map [29].
- Cost-effectiveness criteria. One of the most important criteria for saving expenses and maintaining the environment with the lowest potential so the island does not lose its distinctive environmental character because of human-caused environmental changes. The island was divided into four sectors from west to east, with sector 1 being closest to the port, and sector 4 being the farthest, the aim being to focus urbanization on the western half of the island.
- Land availability criteria. In order to protect archaeological areas and urban areas from human and natural hazards, criteria are selected that provide an urban plan to help preserve historical monuments of the island, especially since the island has several important archaeological sites dating back more than 5000 years [30,31]. For archaeological areas, the island was divided into four archaeological categories (see Land Availability, number 9, Table 1). With regard to the natural hazards, Almatar et al. [15] provided a map of the areas exposed to the danger of sea level rise and sabkhas. This also divided the island into four sectors (Land Availability, number 10, Table 1).
- Socio-economic criteria are divided into two overarching layers: distance from the main town and the productive capacity of the land. Firstly, the island was again divided into four sectors. Secondly, 34 soil samples were analyzed and compared with the results of a study conducted by Abbadi and El-Sheikh [32]. Furthermore, for understanding soil and land use planning in terms of sustainable development, it was also necessary to define the interactions and competitions which exist between the different uses of soil and land [33].

These collected datasets were then integrated into a geodatabase, including many vector feature classes, such as land use highlighting various human activities, as well as the geomorphological map, soil type, and archaeological/historical sites. The vector layers were subsequently employed to create several raster surfaces for further analyses.

**Table 1.** List of criteria and relevant indicators.

| Factors | Weight % | Criteria | Weight % | Rank | | | |
| | | | | 9 | 6 | 3 | 0 |
| | | | | High | Moderate | Low | Unsuitable |
| Environmental | 60 | 1. Elevation (M) | 8 | 10–7.5 | 7.4–5 | 4.9–2.5 | 2.4–0 |
| | | 2. Geomorphology | 15 | Old urban | Flat dry land | Sabkhas | Archaeology site |
| | | 3. Soil PH | 8 | 8.43–8.87 | 8.19–8.42 | 8–8.18 | 7–7.9 |
| | | 4. Coastline type | 15 | Sandy | Sandy rocky | Muddy and gravelly | Rocky |
| | | 5. Geology | 10 | Cemented coastal deposits | Strand line deposit | Dipdipah formation | Sabkhas deposits |
| | | 6. Slope | 4 | 0.93–2.57 | 0.46–0.92 | 0.2–0.45 | 0–0.19 |
| Cost-effectiveness | 15 | 7. Distance to port | 8 | Very near (west) | Near | Middle | Far (East) |
| | | 8. Distance to water and energy line | 7 | Very near (west) | Near | Middle | Far (East) |
| Land Availability | 10 | 9. Archaeological area | 6 | Rest of land | – | buffer zone 400 m | Sites |
| | | 10. Natural hazard | 4 | Zone 1 (6–10 m) | Zone 2 (3–6 m) | – | Zone 3 (0–3 m) |
| Socio-economic | 15 | 11. Land use | 10 | Urban area | Non- urban | Salt deposits | Archaeology site |
| | | 12. Distance from main town | 2 | Zone 1 | Zone 2 | Zone 3 | – |
| | | 13. The productive capacity of the land | 3 | Zone 1 | Zone 2 | Zone 3 | – |

### 3.3. Developing a Composite Suitability Index

For the purpose of this study, a composite suitability index was developed that considered four factors reflecting the four favorable analysis criteria (Table 1). To establish the weighting matrix, 13 criteria were included, with higher values given to overarching landform features, such as geomorphology, coastline type, and geology. The relative weighted values of each factor and criteria were then discussed and reviewed by both the authors and experts from different disciplines in the geosciences (see Table 1 for full discussion).

### 3.4. Selecting Suitable Sites

This step involved calculating a composite suitability index through aggregating various favorable indicators according to their weights. Based on these results, and with little modification (Figure 3), the 13 suitability criteria for the proposed urban plan were further divided into four types: unsuitable, low, medium, and high.

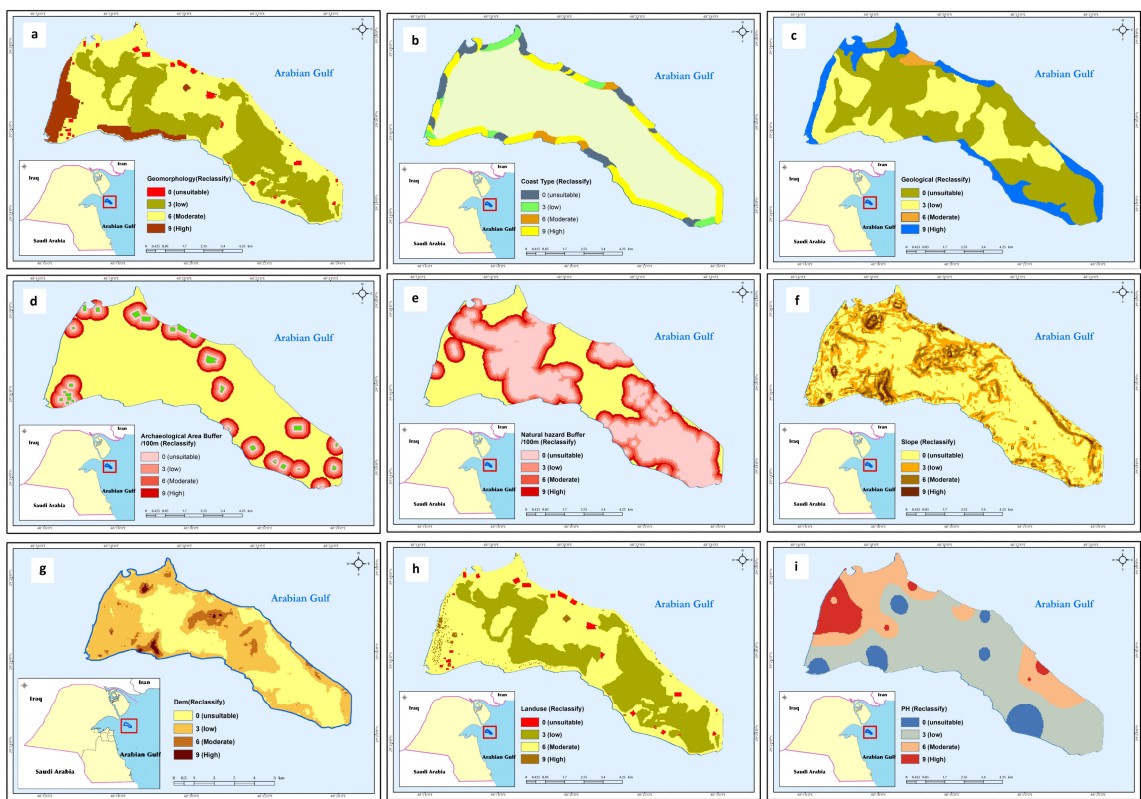

**Figure 3.** Basic layouts of criteria reclassification: (**a**) geomorphology units, (**b**) types of beaches, (**c**) geological formations, (**d**) archaeological sites, (**e**) natural hazard potential, (**f**) slopes, (**g**) digital elevation model (DEM), (**h**) land-use, (**i**) pH.

## 4. Physical and Human Geography of Failaka Island

Examining the physical and human properties of any location is crucial in understanding ecosystems [34]. In this context, the study of climate, geology, soil, geomorphology, history, and archaeological sites for Failaka Island reveals more of the environmental aspects that characterize the island, which in turn contributes to the development of an urban plan based on environmental potential. This potential further contributes to achieving the sustainable development goals (SDGs). Officially known as "Transforming Our World"—the 17 goals set by the United Nations Organization 2015-2016—the 17 Sustainable Development Goals are included in the 2030 Agenda for Sustainable Development, especially goals 11, 14, and 15 [35].

### 4.1. Climate

The island's climate is hot and arid with moderate winds, and the temperature reaches a maximum of 50 °C in summer and a minimum of 4 °C in winter. The average monthly wind speeds on the island are 3.3–7.3 m/s, while the approximate annual rainfall is 125.1 [22,36]. Table 2 displays some climatic characteristics of the study area.

**Table 2.** Climatic characteristics of Failaka Island 2007–2019

| Month | Temp. Ave. (c) | Temp. Min. (c) | Temp. Max. (c) | Relative Humidity Ave (%) | Rainfall Amount (mm) | Wind Speed Ave (m/s) | Wind Speed Max. (m/s) | Evaporation Ave Airport. (mm) |
|---|---|---|---|---|---|---|---|---|
| JAN | 13.5 | 4.6 | 21.7 | 68.4 | 14.9 | 4.6 | 19.1 | 5.4 |
| FEB | 15.9 | 6.5 | 25.4 | 63.3 | 11.9 | 4.8 | 19.9 | 7.2 |
| MAR | 20.4 | 11.1 | 31.7 | 54.7 | 14.3 | 4.8 | 20.8 | 9.8 |
| APR | 25.3 | 15.7 | 37.5 | 49.7 | 6.2 | 4.6 | 24 | 12 |
| MAY | 31.4 | 21.5 | 44.2 | 42 | 5.6 | 4.6 | 20.5 | 15.3 |
| JUN | 35.5 | 25.8 | 48.1 | 30.2 | 0 | 5.7 | 19.1 | 17.8 |
| JUL | 36.7 | 27.4 | 49.2 | 34.5 | 0 | 5.2 | 17.5 | 17.5 |
| AUG | 36.3 | 27.5 | 47.8 | 41.9 | 0 | 4.6 | 15.7 | 16.4 |
| SEP | 33.7 | 23.5 | 45.6 | 42.7 | 0.2 | 4.4 | 16.2 | 14.7 |
| OCT | 28.5 | 18.5 | 41.1 | 52.6 | 10.8 | 4.3 | 19.7 | 11 |
| NOV | 20.9 | 11.7 | 31.1 | 60.6 | 45 | 4.6 | 18.9 | 6.7 |
| DEC | 15.5 | 7 | 24.4 | 66.2 | 16.2 | 4.6 | 17 | 5.2 |

Source: Kuwait Metrological Department [36], 2020. All data from Failaka Island Station except evaporation from Kuwait Airport Station.

### 4.2. Geology

The geological map represents a very important tool for urban planning and, accordingly, understanding geological formations represents an important component for any urban development project. Fortunately, Failaka Island's geological formations are straightforward [18,37] and divided into four categories:

- Cemented coastal deposits or oolitic complexes. An oolitic complex is a series of part marine and part aeolian ridges. These ridges formed alongside Failaka's barrier beaches and coastal dunes, which are separated from the Gulf by coastal lagoons and sabkhas. Likely deposited during high water stands, they consist of oolitic sand, sandstone, and limestone. The area of this formation is 6.9 km$^2$, which constitutes 14.88% of the island's total area.
- Dibdibah formations. These formations, created during the Pleistocene Epoch, appear as coarse-grained, pebbly sand with thin intercalation of clayey sand and clay. These sediments are sometimes combined with calcium carbonate and gypsum, producing conglomerates. They cover 18.81 km$^2$ of the island, or 40.57% of the island's total area.
- Sabkha deposits. Most of these coastal mudflats are widespread on the island, with surface levels varying between 0 and 3 m in height, but being extremely flat and usually barren of any vegetation [29,31].This type of sediment is usually under the constant influence of saline groundwater and susceptible to floods at certain times, often leaving deposits of different evaporite minerals. The sabkha deposits cover an area of 20.1 km$^2$ constituting 43.35% of the total area of the island [14].

- Strand line deposit: These shell-sand and shingle sites appear on the coast in the form of beach gravel and minor unconsolidated beach sand. This geological formation covers the lowest area and percentage of the island (0.55 km$^2$ and 1.17%, respectively), perhaps due to the formation of these sediments on the northern coast, where strong sedimentation streams that formed during rainy periods of the Pleistocene Epoch were washed away by the currents from the Shat al-Arab region when it collided with the Headland of Rouisseyah [29].

*4.3. Soil*

Soils represent a potential source of considerable information in urban planning studies across scales, from specific archaeological sites to broad regional assessments. Applications of soils in the study of the human past include soil geomorphology, the geoarchaeological principles of site formation processes, and the identification of human impacts [38]. Soil surveys are a useful tool for interpreting landscapes, but with the caveat that they were designed for modern planning and land-use studies. Soils are also an integral component of stratigraphic studies at archaeological sites. Owing to these complexities, it was necessary to compare the soil types of the island with the designated plan sites [19]. This endeavor represented one of the most important criteria of our proposed urban plan, as relying on an updated soil map was key to establishing optimum land use planning efforts.

The soil map was prepared based on 34 samples collected in the winter of 2018 covering most parts of the island (Figure 4). Soil study relied on analyses of such criteria as pH, SAR, electrical conductivity, cation and anion exchanges and ratios, and CaCO$_3$ (calcium carbonate), as well as comparing its results with an earlier study by Abbadi and El-Sheikh [39]. Soil production capacity was calculated based on soil pH and cation-anion exchange capacity, and these results, displayed as green areas in Figure 4, were excluded from being urban areas, and instead proposed as agricultural areas within the island plan.

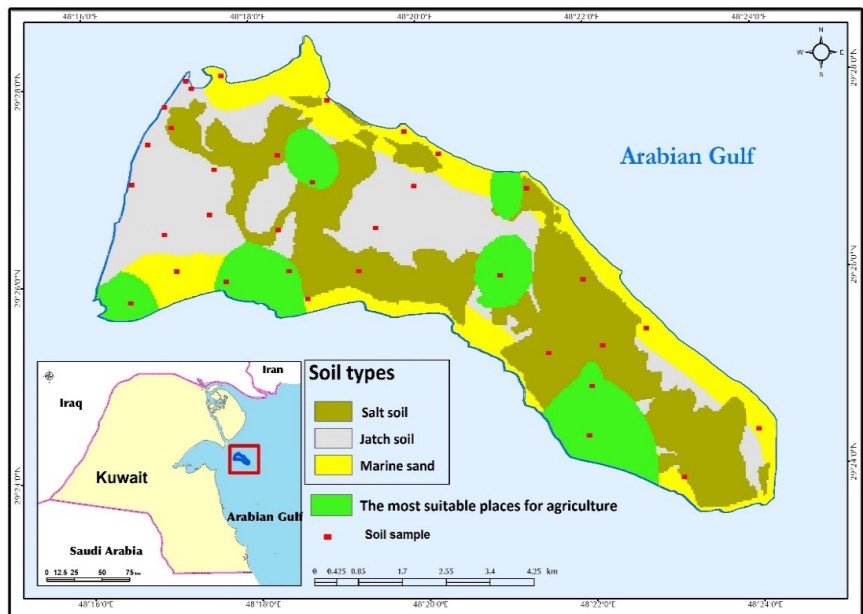

**Figure 4.** Soil types on Failaka Island and the proposed areas for agriculture. Source: [19,39] modified for soil analysis samples procured during this study.

Chemical soil analyses of Failaka Island represent a very important indication of the best arable areas and potential archaeological sites. For example, the detailed role of soil calcium represents an important component of identifying food preparation areas of the past because it is present in the ash of burning charcoal and found in high amounts in teeth and bones. Similarly, middens found in situ represent the source for some of the site's bone and ash disposal areas and contain significant amounts

of calcium [32]. These factors make soil calcium a crucial soil research component in archaeology. Our soil analyses confirmed previous findings, indicating a rise in calcium in sample points 5, 10, 22, 28, and 29, with value amounts ranging from 75–106 (mEq L$^{-1}$).

### 4.4. Geomorphology

During the planning of an urban environment, usually only economic and social parameters are considered [40]. Yet, geomorphology remains an important and effective tool to aid planners and decision-makers in low-cost environmental planning and environmental sustainability [41]. For example, the sabkhas on Failaka Island constitute about 50% of the landmass [15], and therefore building a city or extending a road on the sabkhas areas would cost a significant amount. In this case, understanding the island's geomorphology (i.e., landforms) becomes an effective tool to choose the most appropriate and cost-effective route through the sabkhas, as well as the best location for the city.

Further, the geomorphic processes of erosion and sedimentation processes can be related to the rates of change in the beaches. For example, 68% of the coastline of New England and the mid-Atlantic region of the United States has undergone erosion in recent decades for a variety of reasons, but often related to anthropogenic influences [42]. In contrast, Failaka Island's beaches remain untouched for the past few decades because people have not been allowed to inhabit or visit the island since 1990.

Based on such geomorphic factors then cf., [40,42] the island can be divided into two types of geomorphological forms: those related to the coastline and those on the landmass proper (Figure 5). Indeed, use of the island for touristic purposes cannot be planned without taking the coastline type into account, and thus this study also took into consideration shoreline pattern classification along the island's 38.75 km total coastline length.

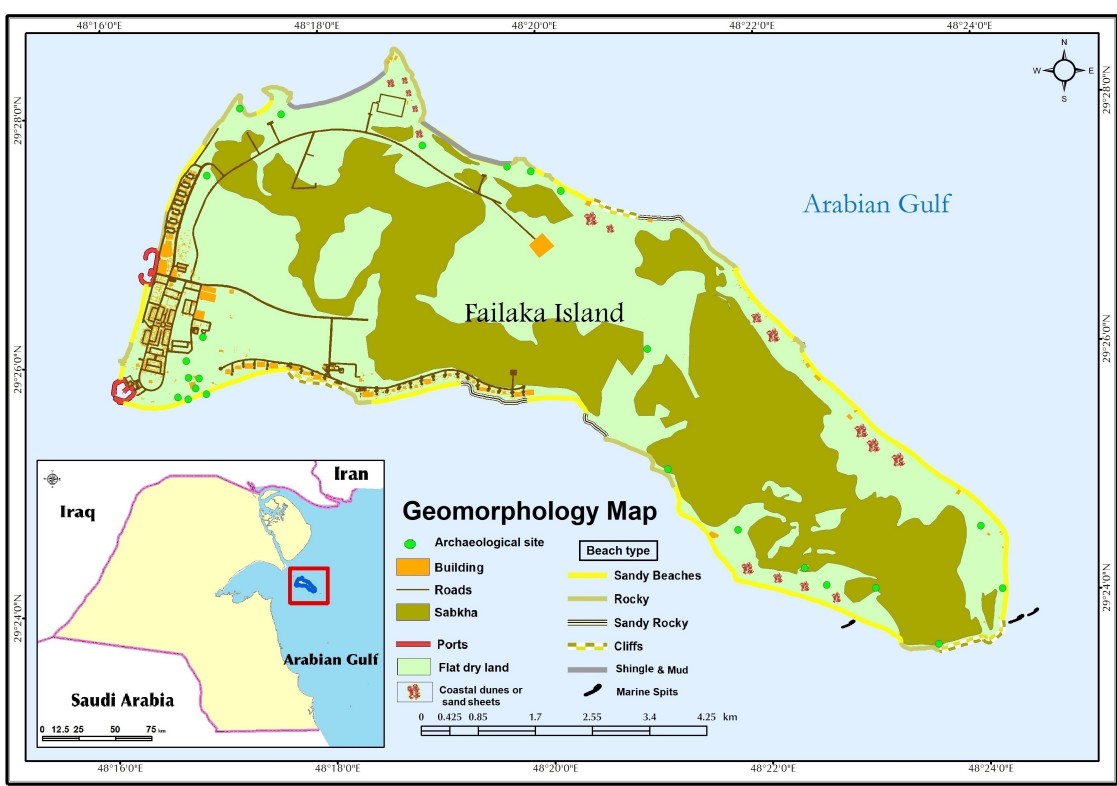

**Figure 5.** Geomorphology map of Failaka Island. Source: Field study and [14].

Coastal landforms can be further classified into two types: erosional landforms and sedimentation landforms (Figure 6). Analysis of satellite images, drone imagery, and field study/ground truthing, reveals five beach types (Figure 5 and Table 3). The classification of beach types remains an essential

pillar of the proposed urban development plan, because the Kuwait government wishes the island to function as a tourist destination. That means limiting the use of sandy beaches for tourism development areas remains an important consideration. As Table 3 demonstrates, half of the island's coasts are sandy beaches, making these areas suitable for urban development and creation of public beaches.

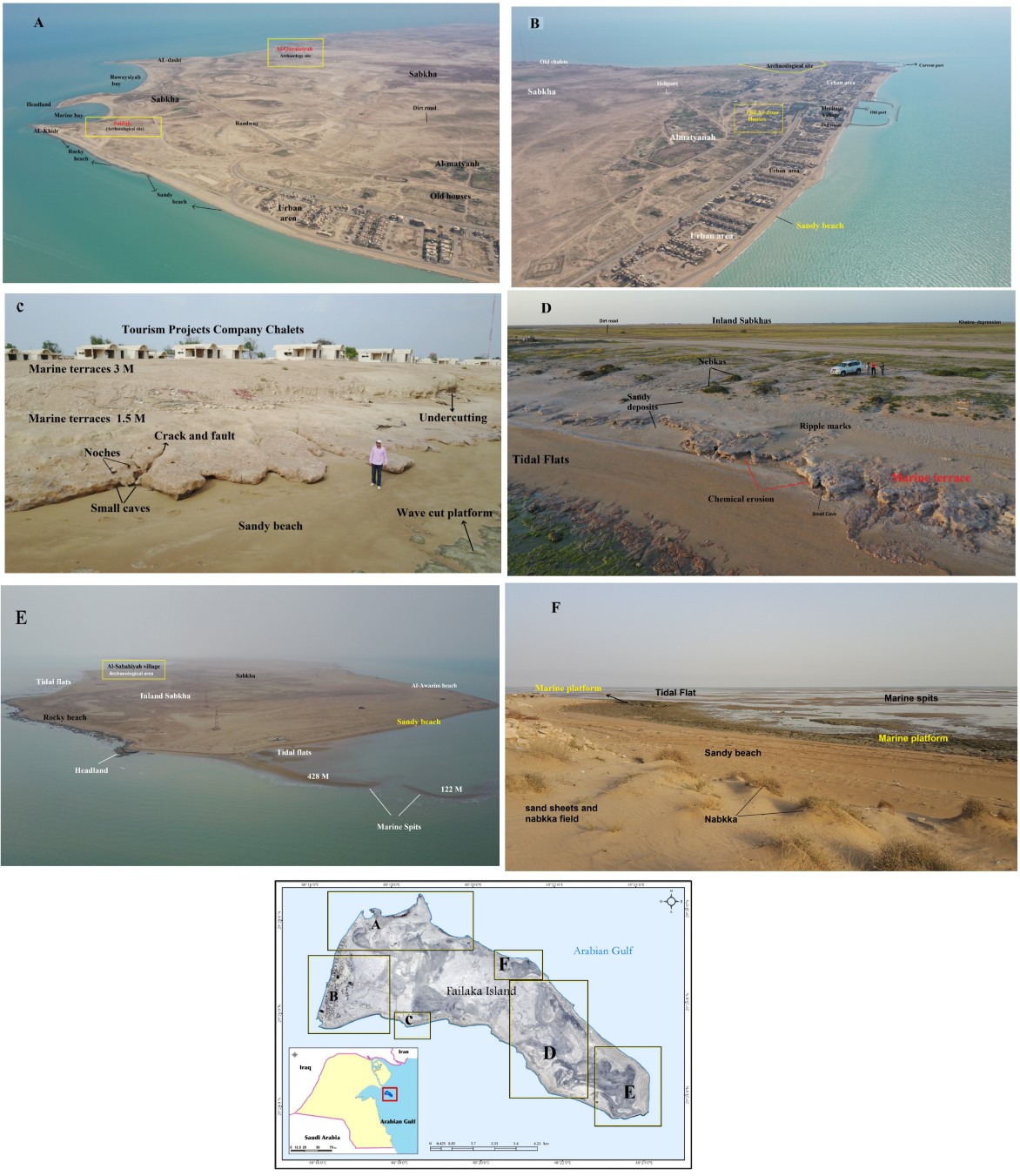

**Figure 6.** Basic geomorphological landforms on Failaka island. (**A**) Marine headlands and bays. (**B**) Sandy beaches and urban area. (**C**) Some of the erosion landforms: Marine cliffs and terraces. (**D**) Inland sabkhas, nebkhas (e.g., coppice dunes), and chemical erosion. (**E**) Marine spits and headlands. (**F**) Sand sheets and coastal sand dunes. Photos captured with Mavic2 pro UAV. A, B, and E from an altitude of 300–456 m and photos C and D from 52 m. Photo E was taken from ground level.

**Table 3.** Lengths of beach types on Failaka Island.

| Type | Length (km) | % |
|---|---|---|
| Sandy beaches | 19.91 | 51.38 |
| Rocky beaches | 9.6 | 24.75 |
| Shingle and Mud beaches | 3.06 | 7.89 |
| Sandy rocky beaches | 2.3 | 5.96 |
| Cliffs | 2.7 | 6.97 |
| Ports | 1.18 | 3.05 |
| Total | 38.75 | 100 |

Source: (Field study/ground truthing, [14]).

Geomorphological processes play an important role in balancing the sediment accumulation/loss as well, given that, between 1976 and 2016, the northern beaches accumulated sand at 77 cm/year, while southern beaches eroded by 78 cm/year [43]. Developing and urbanizing in/near sand dunes or in areas where some other geomorphic danger, such as a foundation being weakened by unstable bedrock, represents potential catastrophe, including loss of life. Due to this, the study sought to understand the geomorphological characteristics of the island, characterized by four anthropogeomorphological units: depressions (sabkhas), flat drylands, previously-built urban areas, and archaeological sites.

### 4.5. Historic and Archaeological Sites

Failaka Island hosted several factors that made it ideal for the establishment of ancient civilizations, including being a strategic location along trade routes, suitability as a natural port, and mediation of other ancient civilizations [30]. According to Bibby [44], the island represents one of the oldest settlements (3000 BC) within marine range of the Arabian Gulf. Bibby [44] also asserts that the island had a distinctive pattern of civilization due to its close contact with neighboring civilizations. Nevertheless, the human settlement of Failaka was not stable throughout history, as it witnessed influxes and exoduses depending on trade activities in the Arabian Gulf [18], as well as periodic epidemics, each of which adversely affected continual habitation on the island. Based on available documents, Failaka civilization can be categorized into three ages: (1) ancient civilization, including the archaeological sites of the Bronze and Iron Age, (2) middle civilizations, represented by the Hellenistic civilization in the southwest of the island, and (3) modern civilizations, including Islamic and Christian civilizations [18,44].

The archaeological sites on Failaka Island extend along the shoreline and through the middle of the island (Figure 7). Most archaeological missions have focused on the southwestern and northwestern parts of the island, and those records support the island's occupation from the late Islamic to the Modern Eras. The Danish archaeological excavations have been operating since 1958, making them the first archaeological excavation in the history of Kuwait [15,45]. Yet, several archaeological sites—such as Al-Dasht, Al-Sabahiyah, Al-Sa'id, Al-Ali, Al-Awazim, and Matitah—have been only surveyed and not studied in any detail. The following sites represent the most important archaeological sites located on the island to date, and would therefore make the most interesting tourism sites.

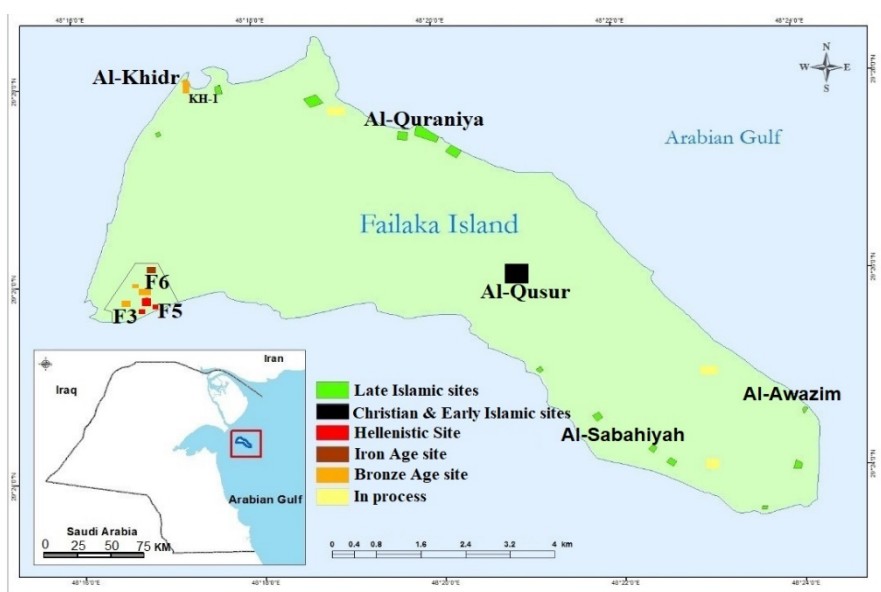

**Figure 7.** Archaeological sites in Failaka island, Source: Field study and [14].

Al-Khidr: A Bronze Age site, known as the Dilmun-Culture port, located in the northwest of Failaka Island. It was fully excavated in 2004. The survey indicated the presence of numerous sherds, stone structures, metal objects, faunal and floral remains, pearls and fishing hooks, and over 600 Dilmun stamp seals. The site is located directly below a modern Islamic cemetery that stretches along the western shore of the shallow Al-Khidr bay (Figure 8c), which served as a port in the past [46,47]. Today, the Al-Khidr site consists of three visible and two less visible mounds.

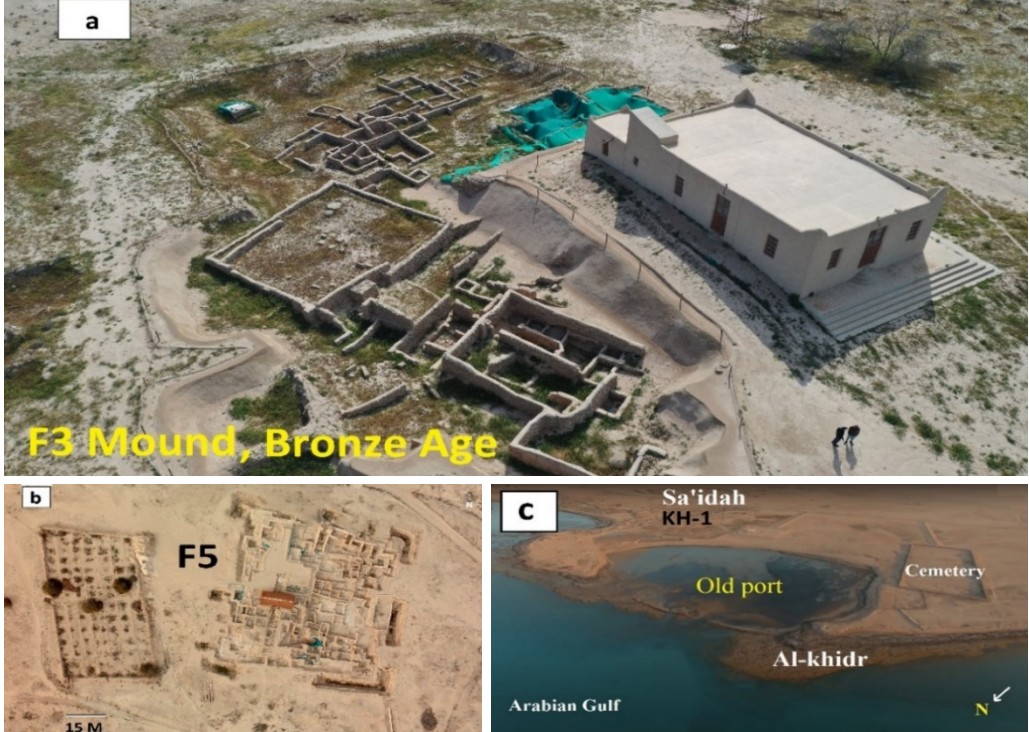

**Figure 8.** Several archaeological sites on Failaka island. (**a**) Oblique photo by UAV, Mavic 2 pro, altitude 41 M, 19 March 2019, (**b**) Mosaic for Hellenistic castle by UAV, Mavic 2 pro, altitude 143 M, 19 November 2019. (**c**) Oblique photo by UAV to Al-khidr, KH-1 and old port.

The KH-1 mound is a low mound roughly 3 to 3.5 m above the contemporary seabed of Al-Khidr bay, stretching 150 m in a north-south direction. It is a sandy dune with shrubby vegetation. Due to high tides, water erosion (2–4 m) has influenced/damaged the growth of its vegetation [46].

F3 mound, a Bronze Age mound referenced by the Danish mission in 1958, and sometimes also known as Tell Sa'ad. It is roughly 9 m above sea level and contains a residential settlement that likely hosted domestic activities, including archaeological evidence of an elite or temple-like structure, the skeletons of gazelles and goats, 170 round stamp seals, and some kilns for an unknown purpose [45]. On the top of this mound, the summerhouse of Ahmad Al-Jaber Al-Sabah, the sheikh of Kuwait, was built during the 1920s [47,48].

F5 mound, also known as Tell Sa'id, is an Iron Age mound that represents the Seleucid culture, post-Alexander the Great. The site was a subject of the first archaeological mission by the Danes (Figures 7 and 8b). It is now understood to be a Hellenistic fortress that consisted of multiple residential units, two temples, multiple storage areas, two gates, and a trench. A French archaeological mission has been working at the site since the 1980s [45].

F6 mound, also known as "the palace", represents a Bronze age mound located near Tell Sa'ad and Tell Sa'id. It is the oldest archaeological site on the island associated with the third Ur dynasty and the Dilmun occupation (~2200 BC). The site consists of a palace-like structure and a Dilmun temple at 4.20 m above sea level [45].

Al-Qusur: Situated in the middle of the island with elevations ranging from roughly two- to five-meters above sea level and surrounded by sabkhas, Al-Qusur represents the largest archaeological site on Failaka Island, covering roughly 2 × 2.6 km [15,45,47]. Over one hundred structures have been found by Italian, Slovakian, and French archaeological missions, and include structural foundations of courtyard houses, church buildings, a central building, and an oval building of unknown purpose. The archaeological records support the presence of a Christian community and a monarchy sometime between the 7th to 9th century AD [15,47].

The Al-Quraniya mound, located in the west-central part of Failaka Island along the north shore, is roughly 550 m long and 250 m wide. With an elevation of roughly 6–8 m above sea level, Al-Quranyia is one of the highest architectural sites on the island, and it contains an old village with 30 structures, mixed grasses and shrubs, and a deserted modern farm [49]. The old village dates to roughly 200–440 years ago, according to the features of the Arabic corner tower, the square courtyard with buildings, the plain and incised pottery, and the glazed ware.

## 5. Results and Discussions

Failaka Island represents one of the most important heritage areas in the State of Kuwait, and there are plans from the state to construct a nature reserve, a residential area, and a tourist marina [15]. Failaka's good environmental condition, freshwater, and fertile soil have contributed to previous initiation of agricultural activities and human settlement as well. Lack of coral reefs on the coastal areas coupled with deep, wide gaps along the coastline have provided for the establishment of several ports on the island in the past ([19] and Figure 8c). Many researchers and archaeological missions concluded that Failaka Island was one of the important cultural centers in the Arabian Gulf region, as it has a distinctive cultural character among other places in the region [15,18,47]. The old general plan of Kuwait [20] mentioned the possibility of exploiting the island for tourism development, with the aim to make it like the Princess Islands (Turkey), Capri (Italy), or the Canary Islands (Spain).

### 5.1. Old Urban Plans

On top of socioeconomic characteristics, physical and environmental factors should also play a fundamental role in the site selection of a development plan. Failaka Island, with no exception, has environmental inputs that cannot be ignored in the site selection processes [19]. While there are multiple other urban plans for Failaka island formulated between 1964 and 2005 (see Figure 9), many

reasons exist that make these plans invalid for implementation. Several examples of why previous urban plans for Failaka Island remain unacceptable include:

- Environmental changes on the island. For instance, an increase in the area of sabkhas and changes in land cover by the length of the island's population abandonment and change in State policy regarding settlement of the island.
- Disregard for important archaeological areas that should to be left empty and protected or developed as part of the tourist or urban areas to generate extra income.
- Spreading of sabkha on the island, which now constitutes 43.3% of the land area, and which should be left empty because of the difficulty in road construction. An intelligent environmental plan will reduce costs to the State rather than resorting to continual soil treatments and raising the level of sabkha, especially considering the effects of sea level rise since 1990, which increased the areal extent of sabkha during that timeframe.
- Ignoring beach type. Even going so far as to put a public beach in the area of rocky beaches, and camping areas in sabkhas ranging in depths from 0–3 m, creating hazardous areas for tourists.
- Lack of intensive fieldwork to survey the island, especially the most recent study, issued by the Municipality of Kuwait, showing that they did not consider the archaeological and environmental components of the island, or several other scientific standards (e.g., soils, geomorphology, prior land use).
- Overlooking the many environmental changes that took place in Failaka in recent years. For example, deserting the island reduced the agricultural areas used, leading to desertified areas where sabkhas have grown to cover 43.3% of the total land area [14,15].
- Being focused solely on the land layout of the island without considering the link between the existing land use patterns and those proposed. For example, it is necessary to compare the soil types (and condition) of the island with the designated site plans using a new soil map of Failaka Island based on recent studies [19,32], and creating a new soil map analysis via soil samples to determine soil production capacity based on total nutrients in the soil.

With regards to inclusion of soils, it should be noted that, although a few of the old plans (cf., [14,19,20,22]( suggested site selection decision was heavily affected by soil fertility, these plans utilized a self-contained city model with agricultural areas that would cover a small population's food needs. The preservation of the agricultural land and allocation of a high proportion of land (50%) of these plans' areas for agricultural utilization is strong evidence of soil fertility's impact on site selection decisions. Even so, the studies focus remained on the most suitable places for agriculture only.

Urban planning based on engineering without considering environmental implications/biodiversity, geoarchaeological sites, and spatial modeling are no longer as suitable as they were in the past. Through spatial modeling based on multiple criteria, it is possible to produce urban plans that achieve the goal of sustainable development, inclusive of any factors influences that ensure preservation of environmental diversity. Additionally, numerous discoveries have occurred at Failaka's archaeological sites over the last two decades, and these should be taken into account when considering any type of plan moving forward.

Given these important reasons, the old urban plans are no longer suitable. Indeed, if those older plans were to be implemented, not only would any prior human footprint be erased by damaging/destroying archaeological areas, but the potential for environmental destruction and geomorphological hazards would also increase. Based on these criteria, there is an urgent need for a new urban plan.

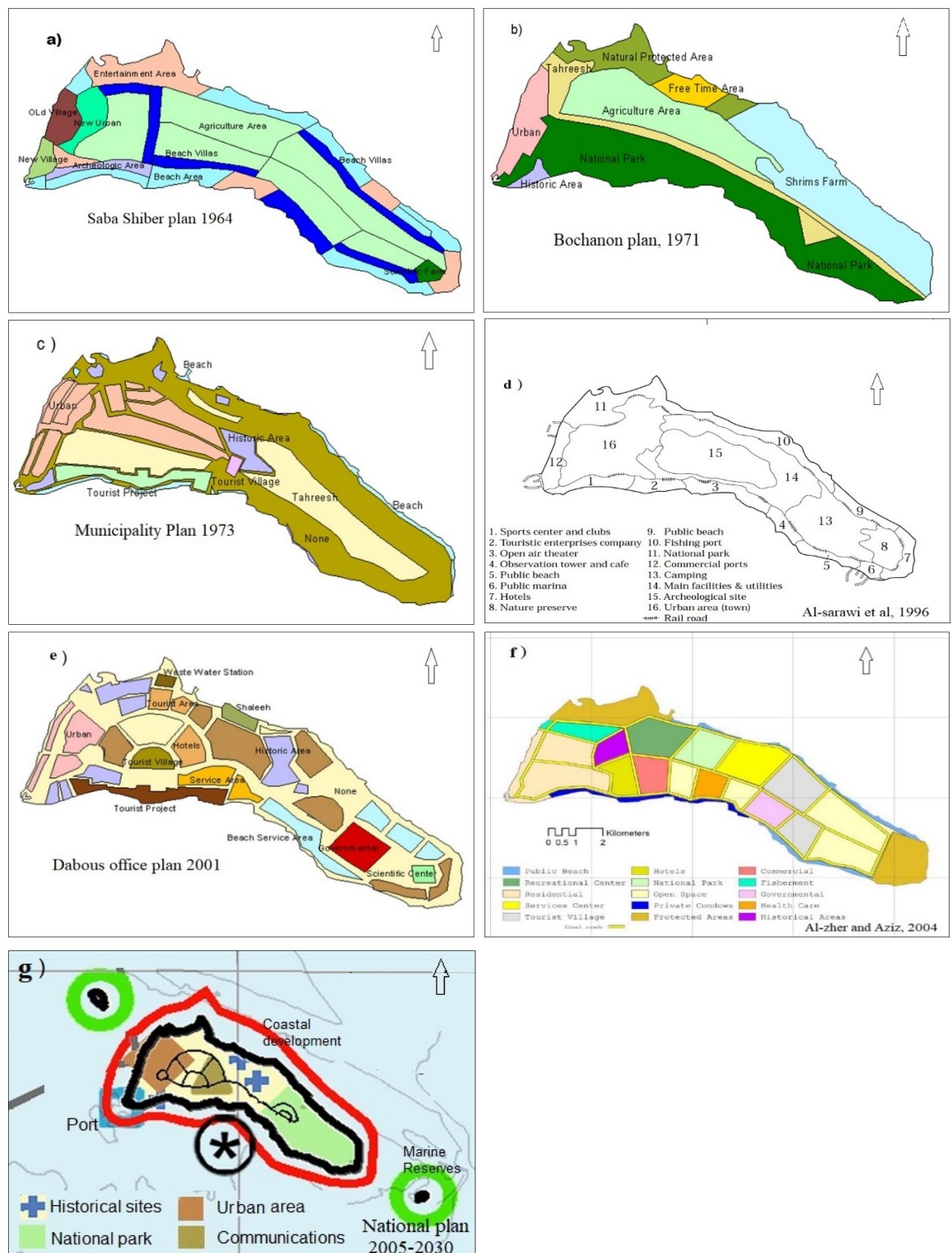

**Figure 9.** Several old urban plans of Failaka island. Source [14,19,20,22].

## 5.2. New Urban Plan for Failaka Island

The proposed urban plan for Failaka Island presented herein is based on environmental factors and natural characteristics, as well as socioeconomic factors. The selection and analysis of sites was dependent on different criteria, as noted in Table 4, and the proposed urban plan utilizes

both high-precision data and high-resolution topographic surveys that relies on three main data sources that produced 13 individual data layers (Table 4): intensive field work, aerial survey by UAV, and high-resolution satellite imagery—all components neglected by previous plans.

**Table 4.** List of data sets used in this study.

| Factors | Criteria | Type | Data Source and Setup |
|---|---|---|---|
| Environmental | Geomorphology<br>Coastline type<br>Geology<br>Elevation (M)<br>Slope<br><br>Soil PH | Favorable | Field work, drone image, World Veiw2, 2010, 2018<br>Field work, drone image, World Veiw2, 2010, 2018<br>KOC,1980<br>Aerial photography, 2004, Kuwait Municipality, 25 cm<br>From DEM<br>Al-zaher and Aziz, 2004; Abbadi and El-Sheikh, 2002;<br>Analysis of 34 soil samples |
| Cost-effectiveness | Distance to port<br>Distance to water and<br>energy line | Favorable | The island divided into four sectors from west to east,<br>where priority was given to urbanization areas in the<br>west than east |
| Land Availability | Archaeological area<br><br>Natural hazard | Boolean | Archaeological areas have been added buffer zone to<br>protect it from urbanism<br>Sabkha area |
| Socio-economic | Land use<br><br>Distance from main<br>town<br><br>The productive<br>capacity of the land | Favorable | The island is divided into four zones, according to the<br>type of land use<br>The island divided into four sectors from west to east,<br>where priority was given to urbanization areas in the<br>west than east<br>Analysis of 34 soil samples |

After correlating the multiple layers, environmental, cost-effectiveness, and socio-economic factors can be considered favorable criteria, while land availability can be considered a Boolean criterion. Upon further analysis of the layers, a set of indicators were identified to reflect criterion in Table 4, and the relationship between each indicator and suitability for a specific urban style that fits each proposed location.

Table 5 and Figure 10 specify the area of each anthropogeomorphological landform on the island and embody an essential pillar for understanding the Earth's surface on Failaka Island. For example, the already-existing urban area represents 5% of the island's total area, which means building on it is very important if the island's heritage sites are to remain intact (and perhaps subsequently used as an income-generating tourist attraction). In the same context, though the current road network is concentrated in the west of the island with a length of about 15 km, it is old, worn-out, and in need of maintenance. However, with a new plan, all new, proposed road networks could be moved away from archaeological areas and geomorphological hazards.

**Table 5.** Area of landforms and percentage of Failaka Island.

| Landforms or Land Use | km$^2$ | % |
|---|---|---|
| Sabkhas | 20.12 | 43.3 |
| Archaeology sites | 1.79 | 3.9 |
| Urban area | 2.34 | 5 |
| Non | 22.19 | 47.8 |
| Total | 46.44 | 100 |

Source: Field study and [14].

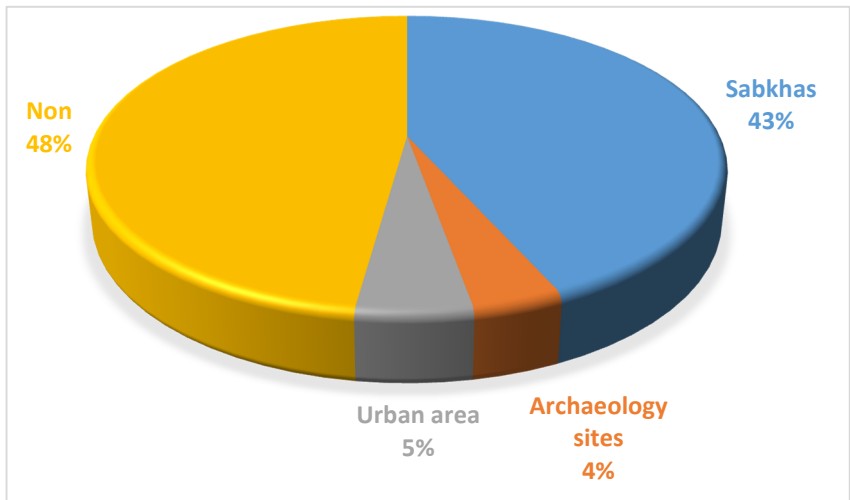

**Figure 10.** Landform percentage of Failaka Island.

Using the EAHP resulted in four suitability ranges: unsuitable, low, moderate, and high. From these data, a suitability map for Failaka island was created (Figure 11). As noted in Table 6, the total area of high suitability for urban use is 6.87 km$^2$ (14.81% of total land area), whereas the total area for moderate urban use is 5.63 km$^2$ (12.1% of total land area). Thus, just over one quarter of the island is potentially suitable for urban development based on the 13 criteria proposed by this study ("High" or "Moderate"). The rest of the island remains unsuitable for development because of the increasing sabkhas and/or rocky/muddy beaches.

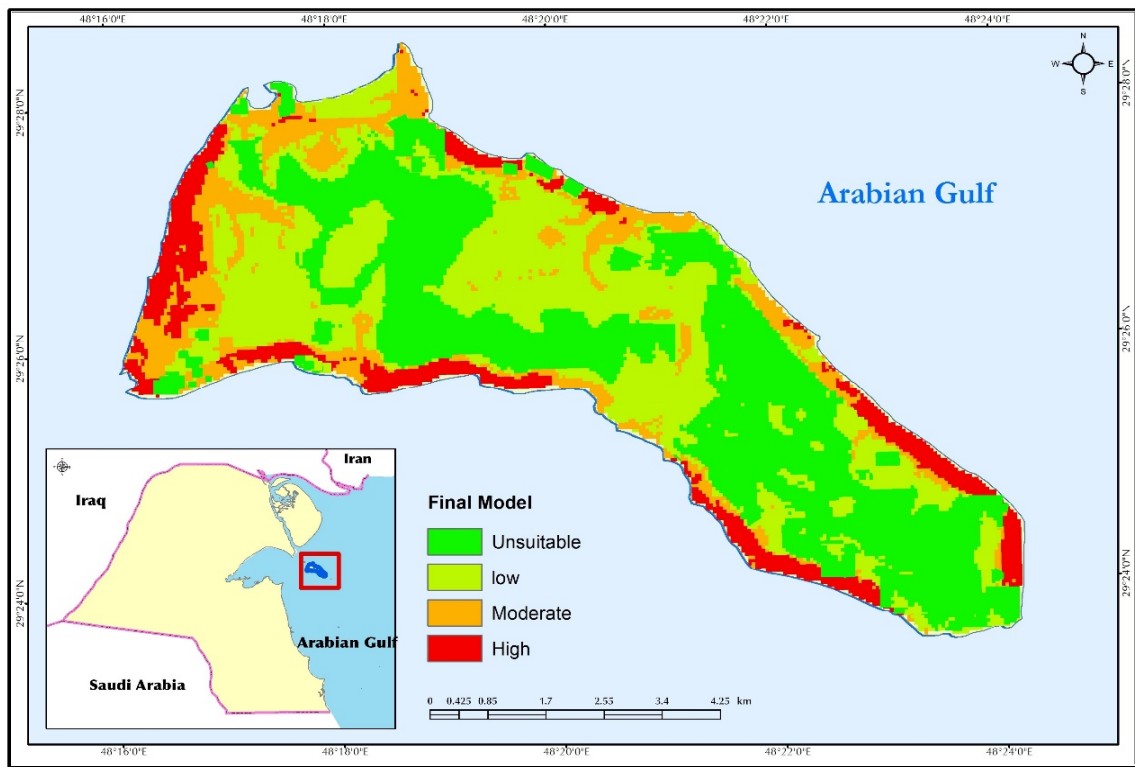

**Figure 11.** The most suitable areas for the proposed urban plan.

**Table 6.** Area and percentage of each category suitable for urbanization.

| Category | Area km$^2$ | % |
|---|---|---|
| Unsuitable | 18.38 | 39.64625 |
| Low | 15.48 | 33.39085 |
| Moderate | 5.63 | 12.14409 |
| High | 6.87 | 14.81881 |
| Total | 46.36 | 100 |

## 6. Conclusions and Recommendations

Urban planning based on environmental criteria represents an essential pillar for achieving sustainable development for future cities. This study suggested the Environmental Analytical Hierarchical Process (EAHP) method and incorporated multiple accompanying factors into a geographic information system (GIS) for further analysis. Out of the 46.36 km$^2$ total land area of Failaka Island, only 12.5 km$^2$, or 27.9%, was found as "suitable" area for development ("High" and "Moderate", see Table 6 and Figure 11). Analysis also showed that sandy beaches of the island represent approximately 51% of the island's total beaches (and about 20 km in length) and, alongside archaeological sites, these remain an essential component for attracting weekend tourists.

Based on years of intense fieldwork by the current research team—as well as previous expeditions and up-to-date modeling and analysis of spatial data—this study suggests several recommendations for moving forward with the planning process of Failaka island as a tourist destination and potential UNESCO World Heritage Site. These include:

- Using transportation that does not pollute the environment, such as a solar-powered electric metro line, circumnavigating the island to avoid the range of Sabkhas and depressions located in the center of the island.
- Preventing vehicles from creating random passages in desert paths, which affects vegetation and living organisms and contributes to non-natural erosion patterns.
- Creating two areas for biodiversity on the island: "Al-Khidr Bay" in the northwest of the island, and "Sabkhas" located in the east of the island between Al-Awazim and Al-Sabahiya.
- Building a road along the coastline that encircles the island (perhaps adjacent to the solar-powered metro line) and two roads that cross the island from north to south, specifically avoiding desert depressions and sabkhas with an elevation of less than 3 m, to avoid the roads turning into salt lakes during the rainy season.
- Enhancing the natural areas of Failaka Island through increased protection and preservation efforts. These natural areas include archaeological sites, coastal dunes, and the coral reef within the intertidal flat.
- Allowing the inland sabkhas to remain in their natural state, using the coastal dunes as a coastal defense in some locations.

Given all the evidence from ground-truthing and GIS analysis, it is clear Failaka Island has environmental components and diversity in its archaeological sites which require Kuwaiti officials and decision-makers to plan in a way that ensures preservation of these factors for future generations—including Kuwait's largest heritage site. Failaka Island's unique physical and cultural diversity has made it a magnet for governmental officials, who want it to become a World Heritage Site, even hosting a recent UNESCO delegation visit. The plan presented herein supports the efforts to designate Failaka island as a World Heritage Site, build a residential city in the west of the island for occasional visitors and a small on-island population, and offer geoheritage and natural tourist attractions, such as the rich archaeological sites and calm, sandy beaches that comprise half of the island's coastline.

**Author Contributions:** Project creation, A.H. and M.G.A.; Funding acquisition, M.G.A.; Methodology, A.H., M.T., and C.D.A.; Project administration, M.G.A.; Writing original draft, A.H. and M.G.A.; Writing—review and editing, A.H., M.T. and C.D.A. All authors have read and agreed to the published version of the manuscript.

**Funding:** This research was funded by [Kuwait University, research sector] grant number [OG01\16].

**Conflicts of Interest:** The authors declare no conflict of interest.

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
