# Peer review of "Environmental Urban Plan for Failaka Island, Kuwait: A Study in Urban Geomorphology"

_sustainability, doi:10.3390/su12177125_

Round 1
Reviewer 1 Report
The work is thorough, well-structured, and its aim is clearly defined and fulfilled. Nevertheless, before its publication, some issues should be solved, particularly regarding format aspects. Mainly, the figures are hardly readable, particularly the text such as legend or scale bar numbers.
Line (L) 40: "In recent years,..." The most recent reference is from 2013, so I would rewrite this part, or some more recent references should be added.
L57: Please, define the acronyms for, unless, GIS and AHP, as later they are used during the text.
L64: Please, add the minutes' symbol, and the second one, to the coordinates.
L66: Maybe, the quotation is too long.
L74: In my view, maybe the sentence "from 1500 to..." could be improved.
L84-85: Please: do not use ".etc" Rewrite the sentence using "such as" or so.
L109: "analyst is provided". This could be a typo.
L114: "analysed" is used in Britain English, and the rest of the text in American En.
L147: In my view, writing "Data preparation represents the first fundamental step...", being the second point (B) of the process (according to the the authors view), is a paradox. A) is Identifying site criteria.
L175: Please, write "5000" instead of 5,000.
L198: Table 1. Rank 3, criteria 2: "sabkhas" should be replaced with "Sabkhas". Rank 0, criteria 6: Please, use the en dash "–", instead of the hyphen "-", and, besides, write "0.19", instead of ".19".
L200-201: Maybe, the sentence could be rewritten.
L225: "3.3-7.3 m/s" should be replaced with "3.3–7.3". Please, authors should make sure if these values are consistent with those from Table 2.
L282: Figure 4: In my view, it would be useful to show the 34 samples' location, if possible.
L294: Please, replace "68%t" with "68%".
L299: Figure 5: The scale bar values are a bit odd. If possible, write better "5 km" than "4.25 km" and so.
L324: In the map, the areas are not easy to be distinguished.
L502-504 and 511-514: Please, revise the format.
L530-531: Please rewrite the sentence to avoid repeating "by the current research team".
Please, revise the style in the whole References Section. Currently, it is very inhomogeneous:
-First name complete / Initial (with/without ".")
-doi/ no doi; "and" or "&" or " ";
-Ref [33] and Ref [35], are the same?
Author Response
Dear sir,
We thank you for your valuable reviews, we will take them into consideration, as they add greatly to our work.
Point 1: Line (L) 40: "In recent years, ..." The most recent reference is from 2013, so I would rewrite this part, or some more recent references should be added.
Response 1: Three new references was added (2017, 2018, 2020).
Point 2: L57: Please, define the acronyms for, unless, GIS and AHP, as later they are used during the text.
Response 2: Done.
Point 3: L64: Please, add the minutes' symbol, and the second one, to the coordinates.
Response 3: it has been modified
Point 4: L66: Maybe, the quotation is too long.
Response 4: Thanks for the good note, I think a 6-line paragraph quoted from 4 references is not a great quote, but it is appropriate, we thank your understanding.
Point 5: L74: In my view, maybe the sentence "from 1500 to..." could be improved.
Response 5: Has been re-wording of paragraph.
Point 6: L84-85: Please: do not use ".etc" Rewrite the sentence using "such as" or so.
Response 6: Done.
Point 7: L109: "analyst is provided". This could be a typo.
Response 7: Paragraph modified and part of it deleted.
Point 8: L114: "analysed" is used in Britain English, and the rest of the text in American En.
Response 8: Modified to analyzed.
Point 9: L147: In my view, writing "Data preparation represents the first fundamental step...", being the second point (B) of the process (according to the authors view), is a paradox. A) is Identifying site criteria.
Response 9: Modified to “an important step”.
Point 10: L175: Please, write "5000" instead of 5,000.
Response 10: Done.
Point 11: L198: Table 1. Rank 3, criteria 2: "sabkhas" should be replaced with "Sabkhas". Rank 0, criteria 6: Please, use the en dash "–", instead of the hyphen "-", and, besides, write "0.19", instead of ".19".
Response 11: Done.
Point 12: L200-201: Maybe, the sentence could be rewritten.
Response 12: The paragraph was rewritten.
Point 13: L225: "3.3-7.3 m/s" should be replaced with "3.3–7.3". Please, authors should make sure if these values are consistent with those from Table 2.
Response 13: Replaced and ensure that numbers are consistent with Table 2.
Point 14: L282: Figure 4: In my view, it would be useful to show the 34 samples' location, if possible.
Response 14: The map has been modified as you indicated.
Point 15: L294: Please, replace "68%t" with "68%".
Response 15: Done.
Point 16: L299: Figure 5: The scale bar values are a bit odd. If possible, write better "5 km" than "4.25 km" and so.
Response 16: Enlarging or reducing the scale bar will affect the size of the display on the island and it will require modification of all the maps and after discussions, the matter was settled on keeping the scale as it is, especially since it is digital and clear.
Point 17: L324: In the map, the areas are not easy to be distinguished.
Response 17: The map has been modified.
Point 18: L502-504 and 511-514: Please, revise the format.
Response 18: The whole paragraph formatting has been modified.
Point 19: L530-531: Please rewrite the sentence to avoid repeating "by the current research team".
Response 19: The paragraph was rewritten.
Point 20: Please, revise the style in the whole References Section. Currently, it is very inhomogeneous: -First name complete / Initial (with/without ".")
-doi/ no doi; "and" or "&" or " ";
-Ref [33] and Ref [35], are the same?
Response 20: The list of references has been revised and updated.
Hope that the revised paper will be up to the standards of journal Sustainability.
Best regards
Ahmed Hassan
Reviewer 2 Report
The article deals with a very specific case study of Failaka Island, located in the Kuwait Bay at about 20 km from the State of Kuwait’s coast. Currently the island receives attention from the state’s institutions as it is “to become the first tourist destination for the State of Kuwait” as a part of the Vision of Kuwait 2035. the authors of the paper review the existing plans for the island prepared in the past and conclude that current situation of the island, as now it has no inhabitants, and the evolving techniques and trends of planning research require new feasibility study and new environmental plan. The authors apply GIS technologies, on site research and other methods to accumulate and analyze the necessary data and propose so-called suitability analysis or suitability plan for the island. The paper is well written, English language is good, the list of references is sufficient; it could serve as an example of suitability analysis case study; however, as the character of the Sustainability journal requires, it would benefit from setting the research into the wider context of sustainability science.
Below I present several questions that the authors could consider:
- Is the scheme in the figure 2 is created by the authors or this is typical course of similar research?
- What is wider research context of this study and it’s scientific novelty; what implications for sustainability science it provides?
- The legends of the maps in the figure 3 are not legible.
- The archaeological data is included into geomorphology section of the article. I would suggest the analysis of heritage values of the island as a separate section as they are very important for the tourism development and the revival of the island in general.
- The authors should specify the term “suitability” in the case of the island: is it suitability for residential urbanization, for commercial, tourism development etc.
- I suggest underlining the ecological, biodiversity aspects more clearly in the analysis as one of the conclusions proposes biodiversity areas in the island.
Author Response
Dear sir,
We thank you for your valuable reviews, we will take them into consideration, as they add greatly to our work.
Point 1: Is the scheme in figure 2 is created by the authors or this is typical course of similar research?
Response 1: Of course, the methodology of the study depends on previous readings and other methodologies in previous studies that have been referred in the element (3.Materials and Methods), but what is new here is the merging between (Environmental Analytical Hierarchical process, EAHP) and (GIS) from the point of view of urban geomorphology. In a neglected area, Failaka Island, which is the largest archaeological site in the State of Kuwait.
Point 2: What is wider research context of this study and it’s scientific novelty; what implications for sustainability science it provides?
Response 2: The methodology used in this study may not be innovative and it is followed in one way or another in several regions around the world. The new! It will be for the State of Kuwait, which is a developing country in which the decision-makers seek to advance the country through urban plans that achieve sustainable development and preserve the environmental heritage. Thus, we can summarize the idea of research in one sentence: "Urban geomorphology and decision-makers."
Point 3: The legends of the maps in the figure 3 are not legible.
Response 3: Figure 3 has been enlarged as much as possible and maybe in the final format before publishing we can change the format or raise the quality of the Figure to be clearer.
Point 4: The archaeological data is included into geomorphology section of the article. I would suggest the analysis of heritage values of the island as a separate section as they are very important for the tourism development and the revival of the island in general.
Response 4: Below this title and on pages 13 & 14 (4.5. Historic and archaeological site). There is a brief explanation of the most important archaeological sites on the island and there is a modern map of all the archaeological sites from 5000 years until now, and because the study is interested in urban geomorphology and the preservation of environmental diversity, after discussions we are satisfied with what was mentioned on pages 13 & 14.
Point 5: The authors should specify the term “suitability” in the case of the island: is it suitability for residential urbanization, for commercial, tourism development etc.
Response 5: Rephrase the last paragraph of (Conclusions and Recommendations) to answer this question.
Point 6: I suggest underlining the ecological, biodiversity aspects more clearly in the analysis as one of the conclusions proposes biodiversity areas in the island.
Response 6: Points 3 and 5 in the (recommendations) refer to this part. Also, throughout the research, there is a focus on the environmental aspects and the heritage diversity in the island, to preserve the island’s environment while developing it, which is the main goal of the research paper.
Hope that the revised paper will be up to the standards of the journal.
Best regards
Ahmed Hassan